# Ultrastructure of the antennal sensilla of the praying mantis *Creobroter nebulosa* Zheng (Mantedea: Hymenopodidae)

**Yuchen Wang[1], Tao Wan[1], Yang Wang[2]\*, Peng Zhao[1], Yang Liu[1]\***

**1** Key Laboratory of Resource Biology and Biotechnology in Western China (Ministry of Education) and College of Life Science, Northwest University, Xi'an, Shaanxi Province, China, **2** Shangluo Research Center of Chinese Medicinal Materials Integrated Pest Management, Shangluo University, Shangluo, Shaanxi Province, China

\* wyang369@163.com (YW); liuyangent@nwu.edu.cn (YL)

**Data Availability Statement:** All relevant data are within the paper and its Supporting Information files.

**Funding:** National Natural Science Foundation of China (Grant no. 32200380); Natural science basic

## Abstract

The praying mantis *Creobroter nebulosa* Zheng (Mantedea: Hymenopodidae) is an insect that has medicinal and esthetical importance, and being a natural enemy for many insects, the species is used as a biological control agent. In this publication, we used scanning electron microscopy (SEM) to study the fine morphology of antennae of males and females of this species. The antennae of both sexes are filiform and consist of three parts: scape, pedicel, and flagellum (differing in the number of segments). Based on the external morphology and the sensilla distribution, the antennal flagellum is could be divided into five regions. Seven sensilla types and eleven subtypes of sensilla were observed: grooved peg sensillum (Sgp), Bohm bristles (Bb), basiconic sensillum (Sb), trichoid sensillum (StI, StII), campaniform sensillum (Sca), chaetic sensillum (ScI, ScII, ScIII), and coeloconic sensillum (ScoI, ScoII). In Mantodea, the ScoII is observed for the first time, and it is located on the tip of the flagellum. The external structure and distribution of these sensilla are compared to those of other insects and possible functions of the antennal sensilla are discussed. The males and females of the mantis could be distinguished by the length of antennae and number of Sgp. Males have antennae about 1.5 times longer and have significantly larger number of Sgp compared to females. The sexual difference in distribution of the Sgp suggests that this type of sensilla may play a role in sex-pheromones detection in mantis.

## 1. Introduction

The antenna of insect is a highly precise sensory structure, it evolved for receiving various environmental signals that are important for different biological and behavioral aspects. The sensilla play vital role in insect life activity, such as habitat selection, food searching, and recognition of sexual partner [1,2]. Previous studies showed that the type and distribution of sensilla in antennae are related to sex, foraging and courtship [3,4]. Structurally, the antenna of mantis is divided into three regions scape, pedicel, and flagellum, the last one is divided into segments (flagellomeres) [5]. Sensilla are the basic structures that allow the perception of environmental

 

research project of Shaanxi Province (Grant No. 2020JQ-904). The funders had no role in study design, data collection and analysis, decision to publish, or preparation of the manuscript.

**Competing interests:** The authors have declared that no competing interests exist.

signals. They have various functions as mechanoreceptors, chemoreceptors, thermoreceptors, and hygroreceptors [6–8].

The praying mantis is a group of predatory insects [9]. They are varied considerably in their ecology, ranging from cursorial hunters to sit-and-wait predators [10]. To detect and attack the prey, the mantids rely on its binocular vision using a pair of the large compound eyes [11]. Both chemical and olfactory cues are also important [12]. Some mantids exhibit characteristic behavioral displays involving colorful patterns on the raptorial fore legs, wings, and thorax [13].

*Tenodera aridifolia* Stoll (Mantodea: Mantidae) is an important and widely distributed predatory insect. In previous studies using scanning electron microscopy of this species, the cuticular fine structure on the antenna was described and the relationship between antennal morphology and behavior was discussed [10,14–16]. The authors illustrated that the predatory behavior of the *T. aridifolia* is affected by the cues coming from sensory sensilla, which determine their food choice [10]. Carle *et al*, developed a new and innovative methodology to reconstruct the antennal development model based on the length of flagellomeres [15]. Six types of sensilla of the *T. aridifolia* were described and illustrated, including grooved peg sensilla, basiconic sensilla, trichoid sensilla, campaniform sensilla, chaetic sensilla, and coeloconic sensilla [16].

The antennal sensilla of only a few species of Mantidae were studied, there have been no study on the species of Hymenopodidae. To reveal the diversity of antennal sensilla, we selected the *Creobroter nebulosa* Zheng, a representative of the family Hymenopodidae, as the research material for the present study. We investigated the morphology, typology, and distribution of the antennal sensilla using light and scanning electron microscopy (SEM). This study attempts to interpret the function of the antenna and provide detailed information on the size, position, and role of the various sensilla. We also compared the characters of antennae of *C. nebulosa* and *T. aridifolia* and described their differences and similarities. We hope that the results of our research may help to understand the sensory function and behavior of *C. nebulosa*.

## 2. Materials and methods

### 2.1. Insect species

Male and female adult mantises (*C. nebulosa*) were collected in Yunnan Province, China. The male is smaller than the female. The male antenna is longer and stronger compared to thinner, shorter antenna of the female. The abdomen of mantis is divided into 9 segments in males and 7 segments in females; the abdomen of females is also wider. More accurate determination of sex can be made by studying the external genitalia. The mantises used in this study were kept in captivity at 26°C and fed with small cockroaches 3–4 times a week.

### 2.2. Optical microscope

Adult antennae of both sexes were observed and removed using an optical microscope (Nikon, SMZ745T, Japan).

### 2.3. Scanning electron microscopy

To identify antennal ultrastructure and their sensillum types, the antennae were removed by cutting the cuticle of the head under the optical microscope. The dissected antennae were cleaned using an ultrasonic cleaner (KQ-250DM, Supmile, kunshan, China) for 5 mins to wash away any particle attached to the surface. After cleaning, the antennae were dehydrated in gradient solutions of ethyl alcohol with the concentration of 50%, 70%, 80%, 90%, 95% and

100% (10 minutes for each). The final dehydration in 100% ethanol was for 30 mins. After dehydration, the samples were dried in critical point dried (SCD-350M, Shianjia, Beijing, China) using $CO_2$ as transfer fluid followed by covering the samples with gold using gold-sputtered for 1 min (GVC-1000, GEVEE-TECH, Beijing, China). The prepared specimens were studied and photographed using a scanning electron microscope (SEM3200, Ciqtek, Hefei, China) with the acceleration voltage set at 10–20 kV.

### 2.4. Data processing and statistical analysis

For our analysis, we took measurements and count the number and distribution of antennal sensilla. The length of antenna is measured by SEM built-in software and the number of sensilla is calculated using Adobe Photoshop CS6 (Adobe, San Jose, CA, USA). Data analysis was completed using analysis of variance (ANOVA) and independent samples t-test by SPSS V. 27.0 software (IBM, Armonk, NY). The results are expressed as the mean ± SE. Values p-less than 0.05 indicate a significant difference and an extremely significant difference. The terminology for the antennal sensilla follows that by of Carle *et al*, and Drilling [5,14].

## 3. Results

### 3.1. Gross morphology of antennae

*C. nebulosa* has filiform antennae that composed of three segments: scape, pedicel, and flagellum (Fig 1). Males have antennae 1.5 times as long as females antennae (Fig 1). The entire width and length of antennae is 12.24 ± 8.61 mm and 165.62 ± 27.99 mm in males (n = 4), and 8.34 ± 7.75 mm and 89.34 ± 21.27 mm in females (n = 4). The proximal 1/3 of flagellum, the flagellomere, of male slightly tapering toward the bottom, the flagellomere of female are not

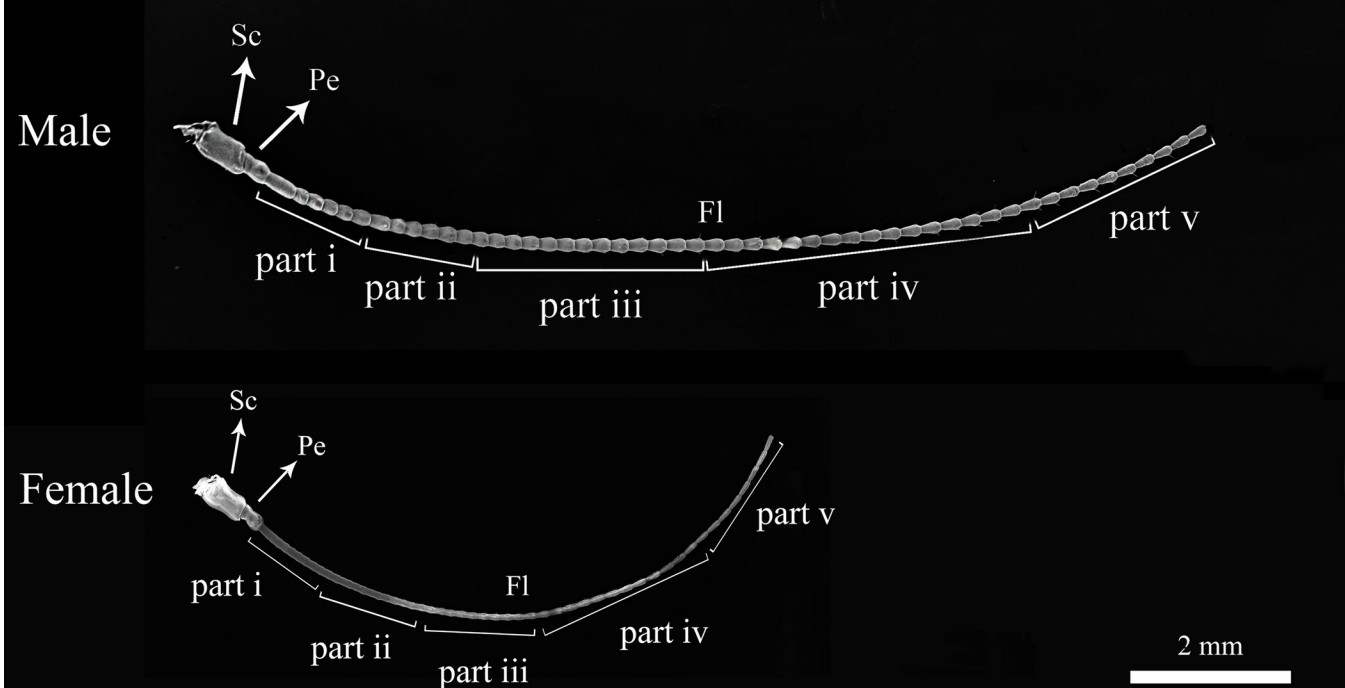

**Fig 1. Scanning electron microscopy images of male and female antenna of the *C. nebulosa*.** Sc, scape; Pe, the pedicel; Fl, flagellum.

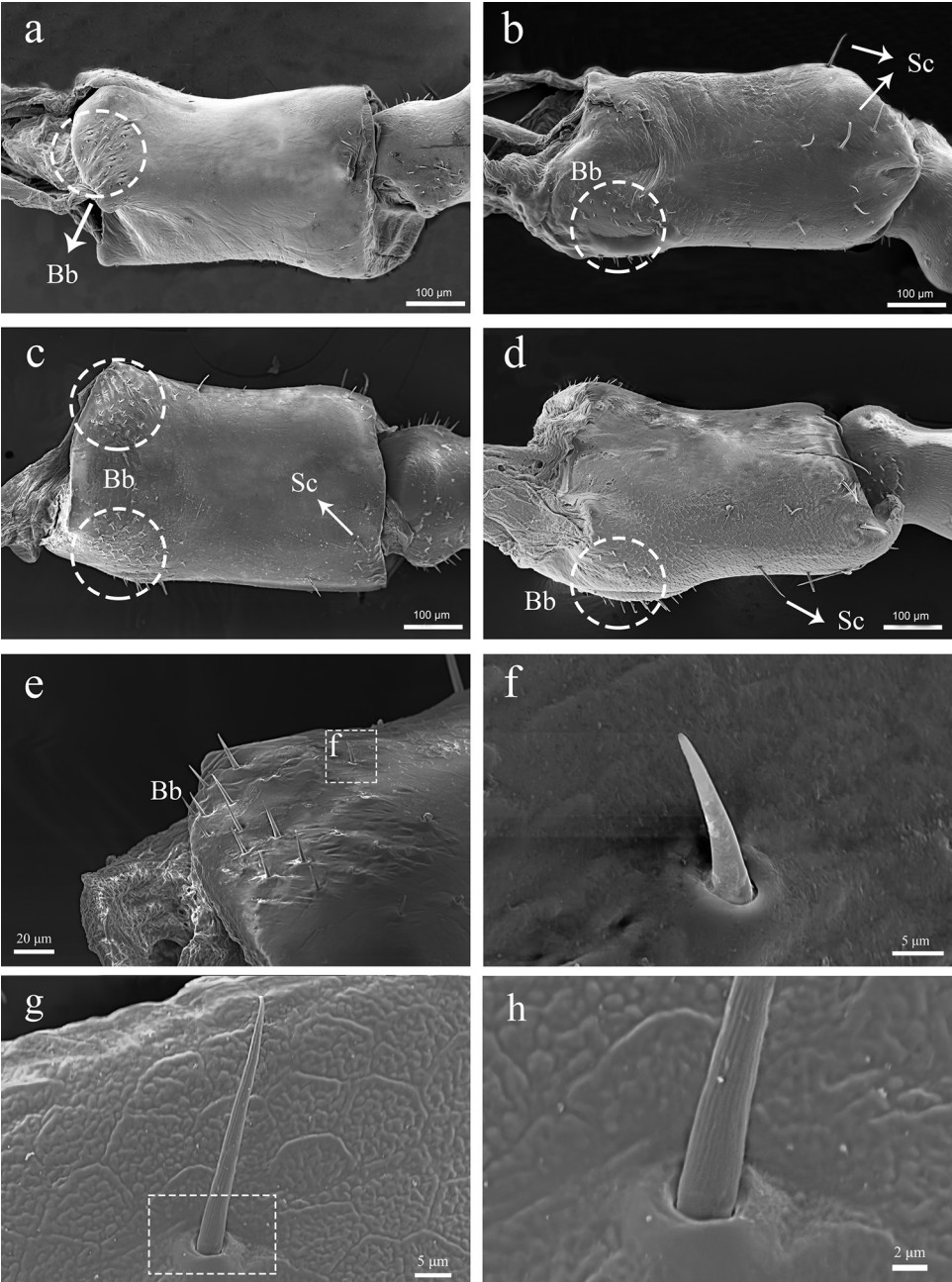

**Fig 2. SEM micrographs showing the right antennal scape of adult male *C. nebulosa*.** (a) Inter lateral view; (b) Outer lateral view; (c) Ventral view; (d) Doesal view. (e) Base of the scape. (f) Bb. (g) Spine-shaped sensilla at the end of the stalk. (h) The base of Sc. Sc, chaetic sensilla; Bb, Bohm bristles.

tapering and have the same diameter through its length. Sensilla of different types are characteristic for different regions of antennae.

The cylindrical scape is connected to the antennal socket where it's functions as the fulcrum point for the rest of antenna. The Bohm bristles are visible on all surfaces except for the ventral surface of the scape base. The number of BB is approximately 20 on the lateral surface (Fig 2A, 2B and 2D), and 50–70 (Fig 2C) on the dorsal surface. A few sensilla chaetica are distributed

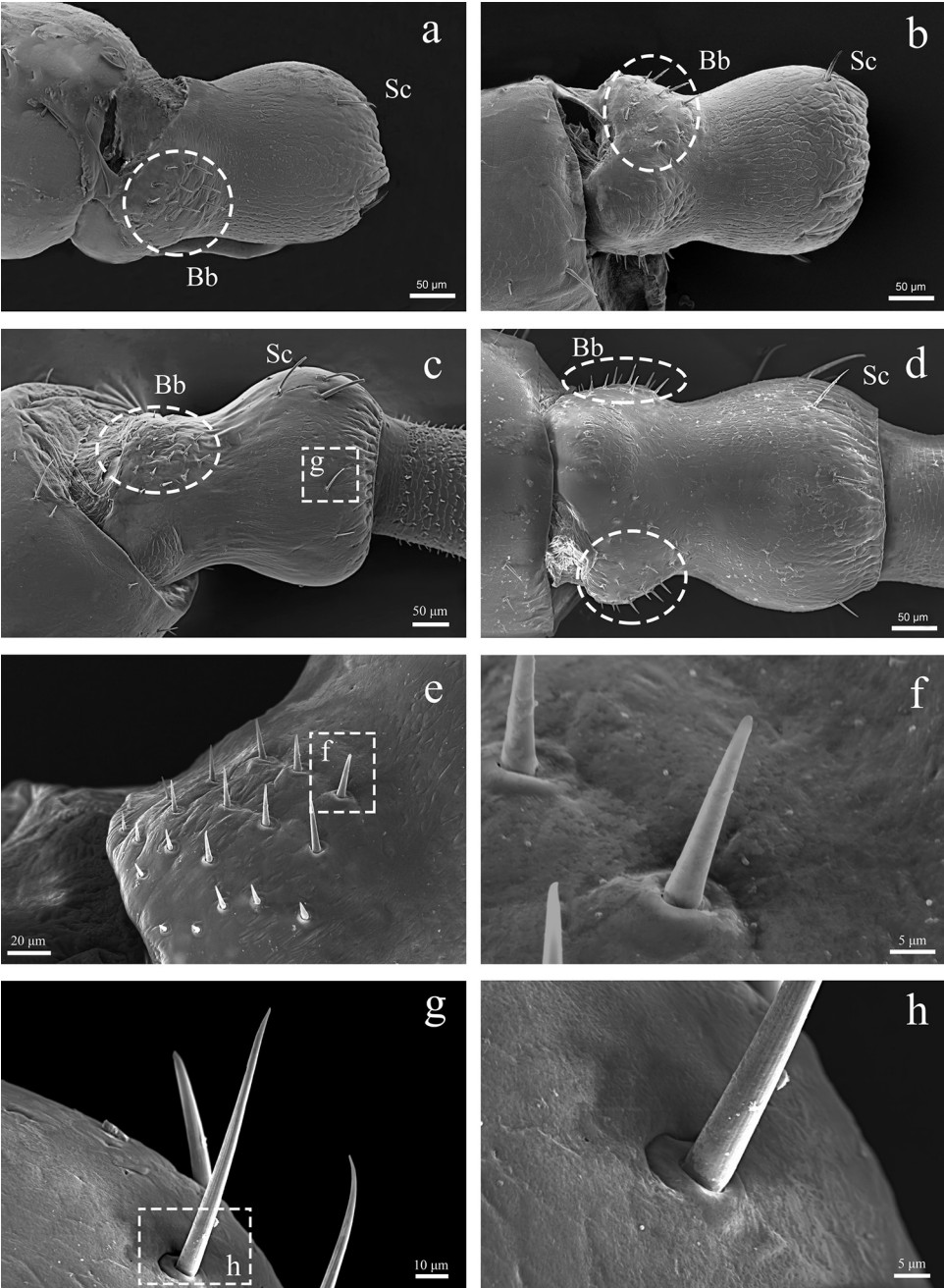

**Fig 3. SEM micrographs showing the right antennal pedicel of adult male C. nebulosa.** (a) Inter lateral view; (b) Outer lateal view; (c) Ventral view; (d) Doesal view. Sc, chaetic sensilla; Bb, Bohm bristles.

on the dorsal and ventral surfaces (Fig 2C and 2D). We did not discover any significant difference in the structure and position of the scape sensilla between males and females.

The pedicel is about 300 μm in length with basal width of 200 μm, and about 150 μm in the middle. It connects the scape and the first flagellomere. Its size is about half of the scape. The type and distribution of sensilla are similar to scape, with a cluster of the Bohm bristles in lateral view (approximately 15–20) (Fig 3A and 3B). The sensilla chaetic are distributed in the

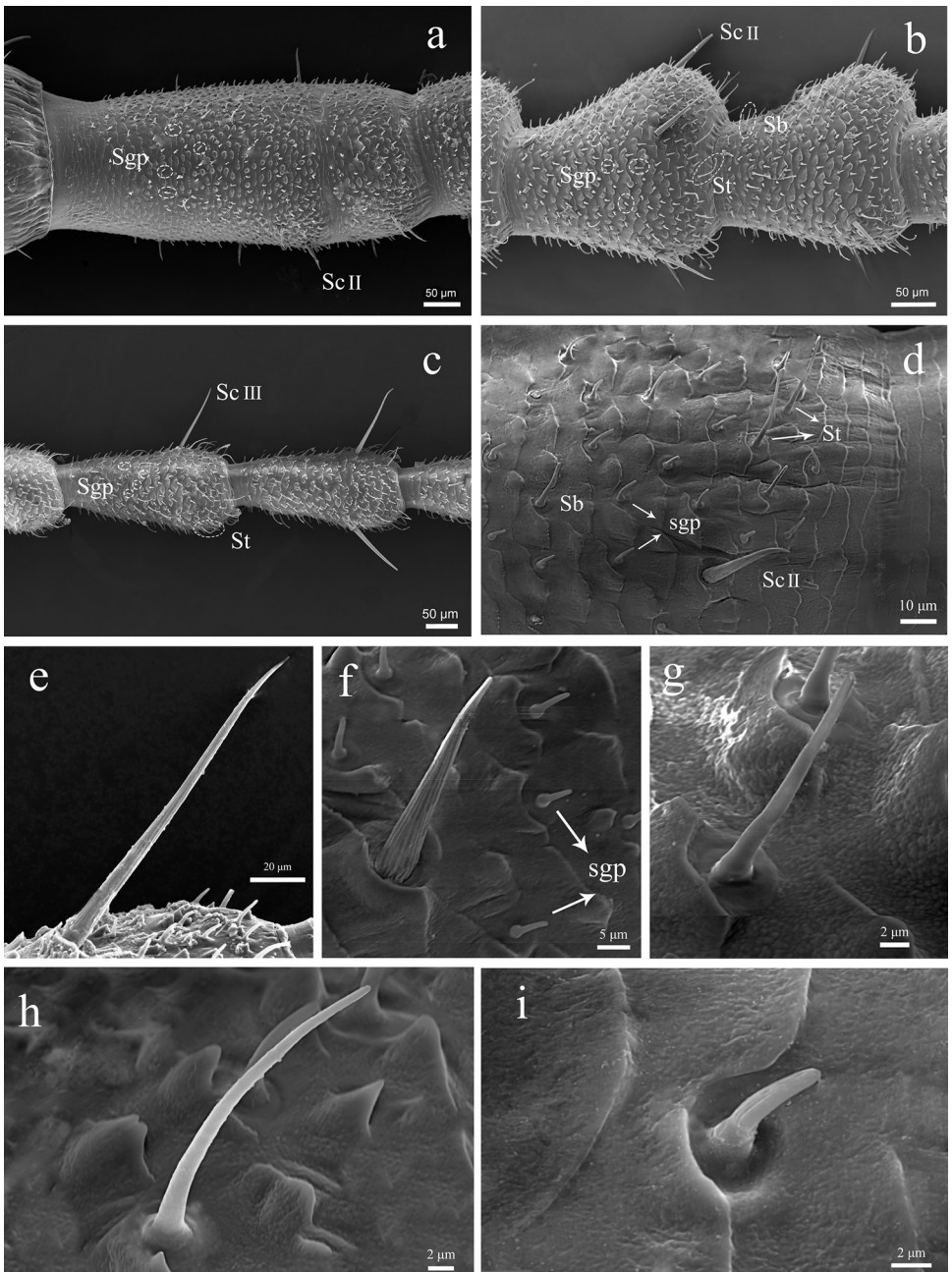

**Fig 4. SEM micrographs showing the flagellum of adult male C. nebulosa right antenna.** (a) The first flagellomere of flagellum; (b) The flagellomeres near the base; (c) The apical flagellomeres. (d) Enlarged view of distal part of the flagellomeres; (e) ScIII; (f) ScII; (g)Sb; (h)St; (i) Sgp. Sb, basiconic sensilla; St, trichoid sensilla; Sc, chaetic sensilla; Bb, Bohm bristles; Sgp, grooved peg sensilla.

distal region (Fig 3A–3D). The length ranging from 45–55 μm and width ranging from 4.5–5 μm. We did not discover significant difference between the male and female pedicel.

The flagellum is the longest antennal segment that consist of 50–60 flagellomeres. The first flagellomere of the flagellum is the longest (Fig 4A), and the second flagellomere is the shortest in both sexes. From the base to the distal part (starting with flagellomere 20–25), the size of flagellomeres gradually increase and they become narrower. Each flagellomere is contracted at

the proximally and become wider distally, slightly narrowing at the point of joint with the next flagellomere (Fig 4B and 4C); The flagellum bears many types of sensilla, such as grooved peg sensilla (Fig 4I), basiconic sensilla (Fig 4G), trichoid sensilla (Fig 4H) and chaetic sensilla (Fig 4E and 4F). The different types of sensilla are characteristic for particular parts of the flagellum.

### 3.2. Description of antennal sensilla

Seven different types of sensilla were observed in both sexes include grooved peg, Bohm bristles, basiconica, trichodea, campaniformia, chaetica, and ceoloconica.

**3.2.1. Grooved peg sensilla.** The grooved peg sensilla are short with significant differences between males and females (Table 1). In a male, the grooved peg sensillum is 8–11 μm in length and 2–3 μm in width, while sensillum in a female is 5–10 μm in length with a diameter

**Table 1. Lengths and basal widths of different types of sensilla of *C. nebulosa*.**

| antennal segments | sensillum type | Length (μm) | | t-Test | Basal Width (μm) | | t-Test |
|---|---|---|---|---|---|---|---|
| | | Males | Female | | Males | Female | |
| **Scape** | Bb | 12.67±2.24 | 15.35±5.88 | * | 2.44±0.28 | 2.68±0.69 | NS |
| | ScI | 65.07±15.71 | 55.42±17.42 | NS | 5.46±1.18 | 5,34±0.91 | NS |
| **pedicel** | Bb | 17.18±2.59 | 19.78±4.12 | NS | 3.93±0.50 | 3.44±0.31 | NS |
| | ScI | 62.56±22.83 | 53.80±9.31 | NS | 5.39±1.26 | 5.20±1.49 | NS |
| | Sca | Present | Present | / | Present | Present | / |
| **Flagellum** | **Part i** | | | | | | |
| | ScII | 44.91±2.53 | 30.45±1.46 | * | 4.08±0.72 | 3.08±0.14 | * |
| | Sgp | 8.17±1.40 | / | / | 2.17±0.16 | / | / |
| | **Part ii** | | | | | | |
| | ScII | 65.62±10.59 | 44.27±10.01 | * | 7.04±0.71 | 4.51±0.86 | * |
| | Sgp | 8.75±1.32 | / | / | 2.65±0.47 | / | / |
| | **Part iii** | | | | | | |
| | ScIII | 118.43±9.97 | 53.08±5.43 | * | 7.98±0.64 | 5.20±0.35 | * |
| | Sgp | 9.56±0.81 | 5.65±1.04 | * | 2.70±0.26 | 2.18±0.11 | * |
| | StI | 25.53±1.96 | 24.08±2.01 | NS | 2.51±0.48 | 2.04±0.12 | NS |
| | StII | 29.47±1.22 | 24.10±3.34 | NS | 2.23±0.24 | 2.11±0.16 | NS |
| | Sb | 13.59±1.83 | 15.75±1.24 | NS | 2.25±0.76 | 2.29±0.25 | NS |
| | **Part iv** | | | | | | |
| | ScIII | 141.3±14.87 | 54.26±2.75 | * | 8.43±0.35 | 4.67±0.58 | * |
| | Sgp | 10.85±1.78 | 7.45±1.36 | * | 3.19±0.49 | 2.57±0.33 | * |
| | StI | 29.41±1.33 | 31.51±2.19 | NS | 2.66±0.28 | 1.97±0.29 | * |
| | StII | 35.64±2.53 | 32.53±3.78 | NS | 2.25±0.12 | 2.11±0.31 | NS |
| | Sb | 17.56±1.16 | 19.35±2.23 | * | 2.13±1.48 | 1.25±0.78 | NS |
| | **Part v** | | | | | | |
| | ScIII | 143.45±15.80 | 69.27±6.82 | * | 7.43±0.18 | 5.79±0.48 | * |
| | Sgp | 10.02±1.82 | 10.32±2.25 | NS | 2.66±0.24 | 2.31±0.27 | * |
| | StI | 31.78±9.08 | 32.43±4.23 | NS | 2.66±0.22 | 1.85±0.11 | * |
| | StII | 47.35±8.17 | 39.46±2.50 | NS | 2.31±0.07 | 2.04±0.19 | * |
| | Sb | 21.42±2.35 | 22.65±1.11 | NS | 1.75±0.72 | 2.55±0.64 | NS |

**Note:** Data are presented as Mean ± SE (n); n, sample size; NS, nonsignificant differences

* indicates p < 0.05 in the independent samples t-Test. Abbreviations: Sb, basiconic sensilla; Sc, chaetic sensilla; St, trichoid sensilla; Sgp, grooved peg sensilla; Sca, campaniform sensilla.

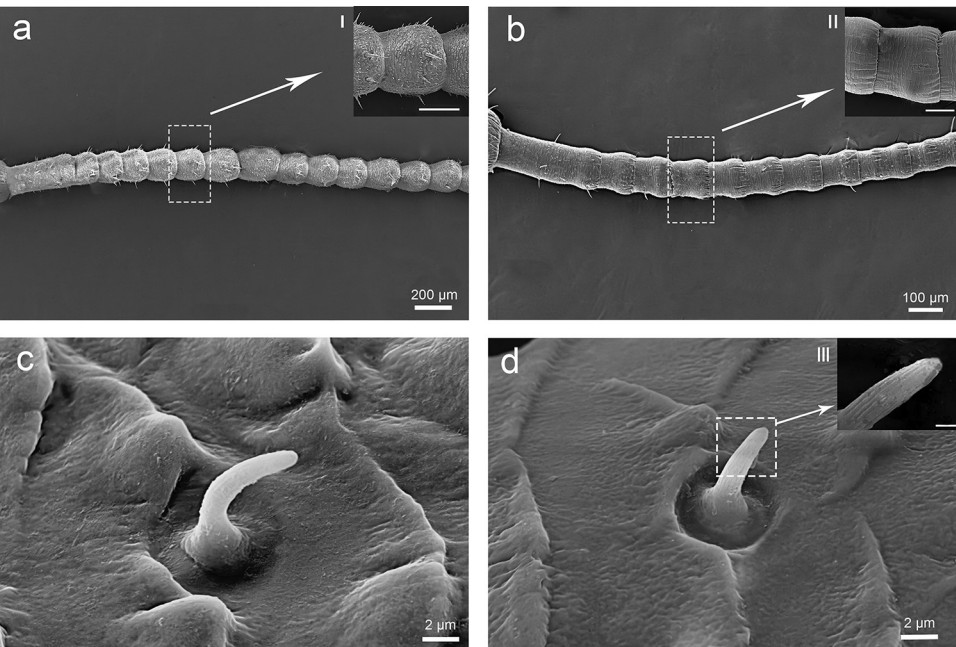

**Fig 5. Distribution and details of grooved peg sensilla.** (a) Distribution of male grooved peg sensilla on the flagellum. (b) A Distribution of female grooved peg sensilla on the flagellum. (c,d) Different external shapes of grooved peg sensilla on the flagellum. Scale bars: I = 100 μm, II = 80 μm, III = 1 μm.

of 2–2.5 μm (Fig 5C and 5D). The grooved peg sensilla distributed on all surfaces of the entire flagellum in males (Fig 5A), but only individual grooved peg sensilla appear on the female antenna at least after the 20th node of the flagellum, gradually increasing in number towards the distal antennal flagellomeres (Fig 5B). The outer surface of the cuticle is sculptured with longitudinal grooves along most of antennal length and has a smooth surface at its proximal part. This sensilla is straight or slightly curved (Fig 5C and 5D). There was no significant difference in the shape of male and female grooved peg sensilla.

**3.2.2. Bohm bristles.** The Bohm bristles occurred at the base of scape and pedicel. The BB have a smooth, non-porous surface, and a thicker base (Fig 6E). The BB is directed nearly perpendicular to the surface of scape and pedicel, and emerges from a slightly raised socket which tightly embraces the base of the shaft. In males, the BBs are 15–20 μm in length and 2.5–4 μm in width. In the females, they are 12–17 μm in length and 2–4 μm in width.

**3.2.3. Basiconic sensilla.** The basiconic sensilla are hair-like structures which are slightly longer in females. In male, their length ranging from 30–45 μm and width ranging from 2–2.5 μm; the female they are 25–40 μm in length and 2–2.2 μm in width. Sb have a smooth perforated surface and a non-flexible socket (Fig 6M), no significant difference in shape was found between male and female.

**3.2.4. Trichoid sensilla.** Trichoid sensilla are long, hair-like, with smooth surface. They usually start in 20–25 flagellomeres. The length of sensilla is gradually increase towards the tip of antenna, up to about 50 μm. Based on their external morphology, we divide the trichoid sensilla into two subtypes. StI are curved or straight, the length is 25–50 μm; the base of StII is intumescent (Fig 6L), the length is 20–30 μm. There was no significant difference in the length of StI and StII between males and females.

**3.2.5. Campaniform sensilla.** The campaniform sensilla have a dome-like structure. We found that some of Sca have obvious ridges on the edges of both sides (Fig 6C), and some

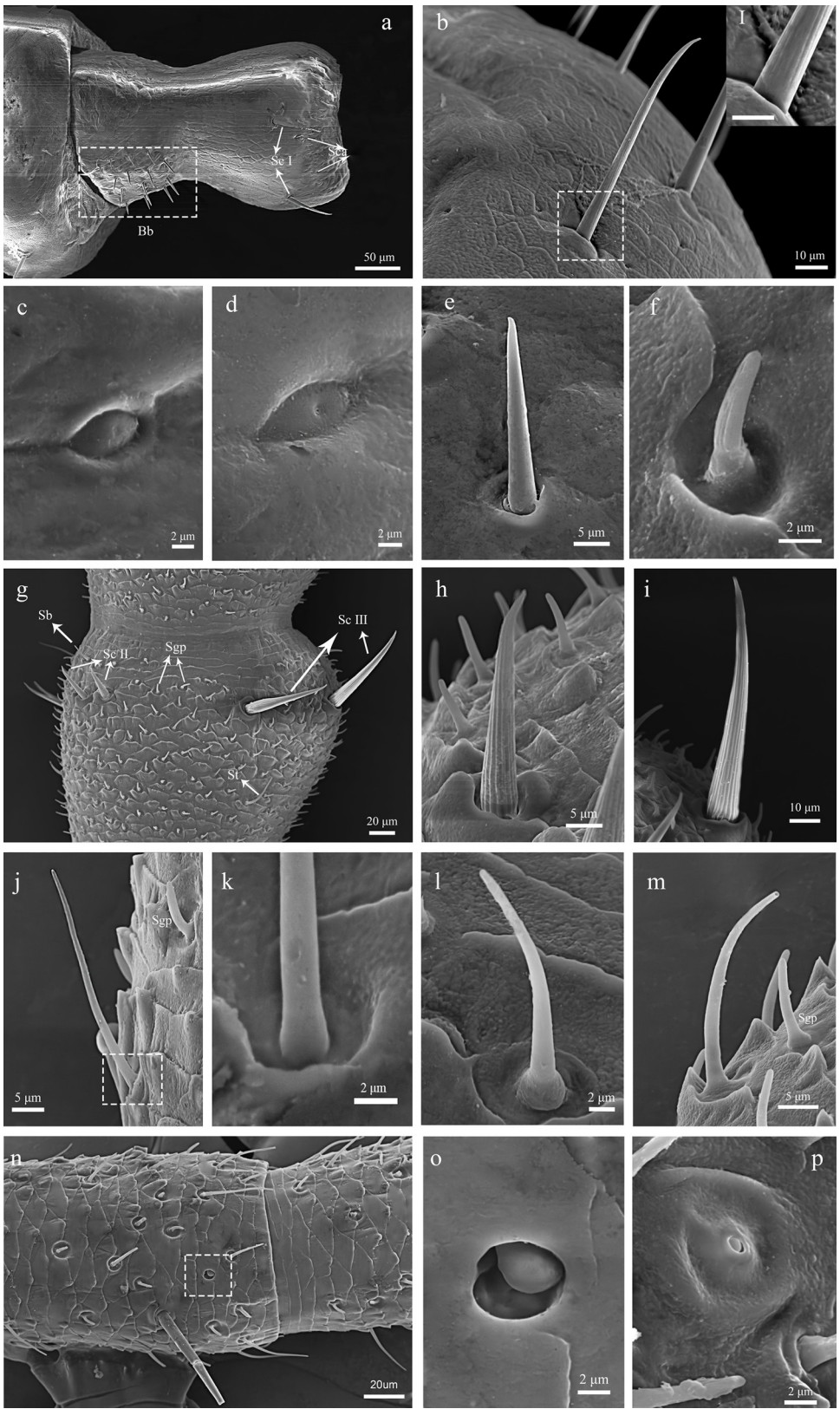

**Fig 6. Different sensilla types on the antennae of adult C. nebulosa.** (a) pedicel. (b) ScI. (c) Sca. (d) Sca. (e) Bb. (f) Sgp. (g) flagellomere. (h) ScII. (i) ScIII (j) StI. (k) The base of StI. (l) StII. (m) Sb. (n) flagellomere. (o) ScoII. (p) ScoI.

Sc, chaetic sensilla; Sca, campaniform sensilla. Bb, Bohm bristles; Sgp, grooved peg sensilla; St, trichoid sensilla; Sb, basiconic sensilla; Sco, coeloconic sensilla; Scale bars: I = 5 μm.

others have smooth edges (Fig 6D). The entire of structure of Sca is about 8–8.5 μm in length and 4–5 μm in width, with a hole of 0.5–1 μm in the diameter in the middle, no significant difference in the length of Sca was observed between males and females.

**3.2.6. Chaetic sensilla.** The chaetic sensilla are the longest sensilla found in *C. nebulosa*. The sensilla are inserted in a socket (Fig 7B, 7H and 7I), and have grooved surface (Fig 7C and 7D). ScI is a slender hair-like sensilla, 50–150 μm in length and 4–5 μm in width, straight or curved, strongly pointed apically. It has indistinct longitudinal grooves (Fig 7B), and distributed in the distal part of the scape and pedicel (Fig 7A); ScII is short and thick, like a spike (Fig 7H), 40–65 μm in length and 7–8 μm in width, bearing distinct grooves along their entire circumference, distributed on flagellomeres near the base. ScIII is longer than ScII, its longitudinal grooves are also evident (Fig 7I), distributed in the distal part of each flagellomere (Fig 7J), its length varies greatly, gradually increasing from the base to the tip of the flagellum, up to about 150 μm.

**3.2.7. Coeloconic sensilla.** The coeloconic sensilla possess a grooved cuticular apparatus protuberance, which stands in a round hollow of 7 μm in diameter rimmed by a cuticular edge; only the tip of the sensilla is visible externally (Fig 7F); there are few coeloconic sensilla on the flagellum. A possible subtype with a larger central apparatus appears on the flagellum (Fig 7O).

**3.3. Sensillar distribution.** The sensillar distribution pattern varied along the antennal longitudinal axis and between the sexes. To described the variations precisely, we developed a new terminology to reference the distinct parts of the flagellum. Instead of using the traditional terms proximal, medial, and distal, which refer to imprecise longitudinal localization. We divided the flagellum into 5 parts from the most proximal (part i) to the most distal (part v) (Figs 1 and 7). Each part has the following characteristics.

**3.3.1. Part i (flagellomeres No.1 to No.8/10).** In the part i, the chaetic sensilla are distributed along a single circular line in the distal region of each flagellomere in both sexes. There are differences were observed between the sexes: in male, grooved peg sensilla are present on the first flagellomere of the flagellum, densely distributed on all sides (Fig 7A). Such sensilla were not observed in females in the part i (Fig 7B).

**3.3.2. Part ii (flagellomeres No.8/10 to No.15/20).** The length of flagellomere is gradually increasing, but it reduced at the flagellomeres No.8 to No.10 (Fig 7C and 7D). Based on this characteristic, we referred to flagellomeres No.8/10 to No.15/20 as the part ii. The length and width of the flagellomeres decreased by about 50 μm and 15 μm respectively in both sexes.

**3.3.3. Part iii (flagellomeres No.15/20 to No.30).** Trichoid sensilla began to occur in this part in both sexes and grooved peg sensilla began to appear in females (Fig 7E and 7F); we use this as a criteria to distinguish the part iii from the part ii. The grooved peg sensilla are the most abundant in male, and the trichoid sensilla are the most common in females in this part.

**3.3.4. Part iv (flagellomeres No.30 to No.40/45).** In this part, the number of trichoid sensilla increases; the sensilla are distributed almost throughout the flagellum in both sexes (Fig 7G and 7H), and the length of sensilla slightly increases towards the distal part of antenna. The number of grooved peg sensilla significantly increases, and they are distributed in two thirds of the flagellomeres in females. (Fig 7H). The chaetic sensilla significantly increase in the length. In the part iv, the length of the chaetic sensilla is significantly different in both sexes, reaching 141.3±14.87 μm in males and 54.26±2.75 μm in females.

**3.3.5. Part v (flagellomeres No.40/45 to the distal end).** Although, the grooved peg sensilla are dominant on the flagellomeres from the part i to iv in males, both sexes display a

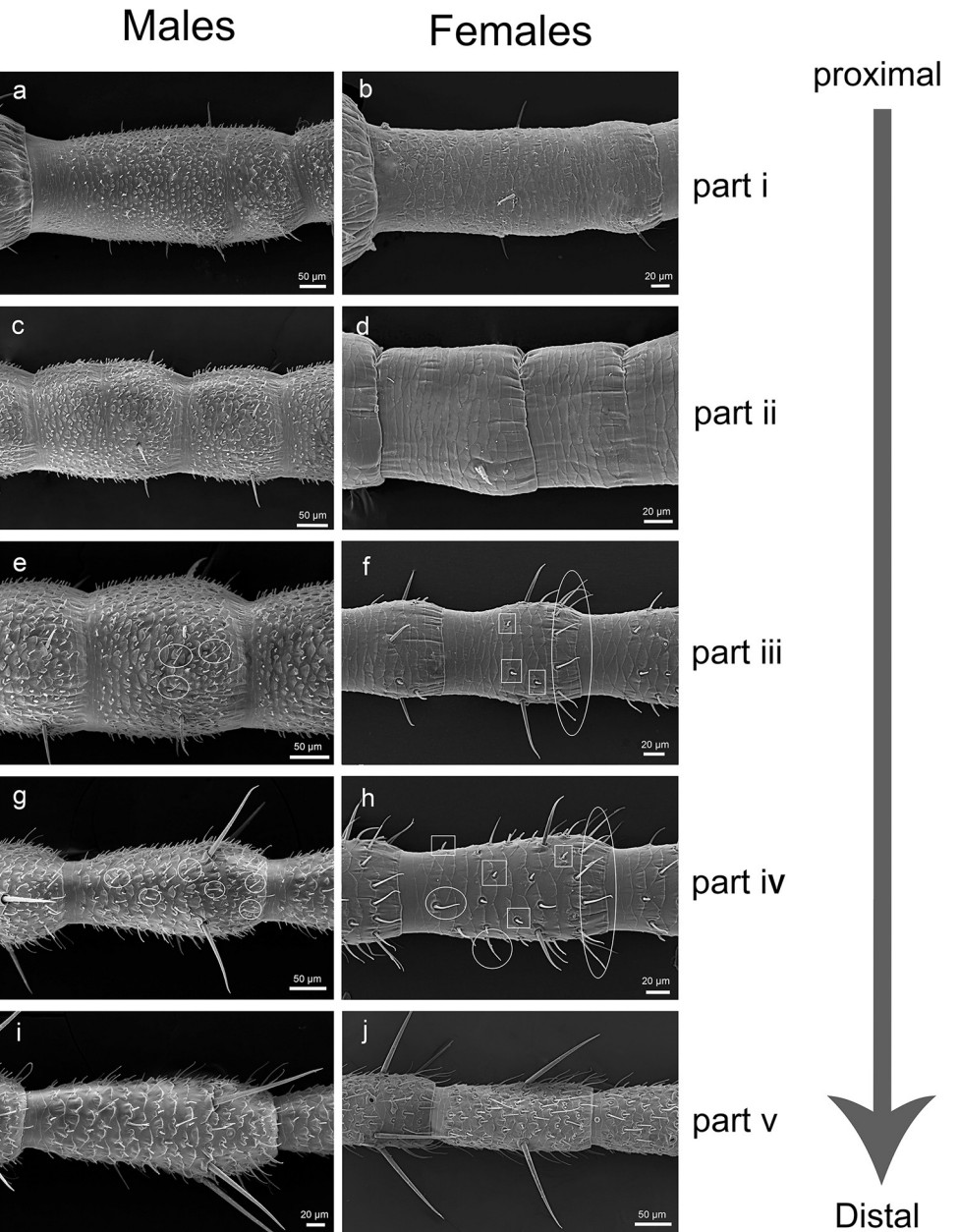

**Fig 7. Variation and sexual dimorphism in sensillar distribution on the flagellum.** (a) The first flagellomere of male C. nebulosa in part i. (b) The first flagellomere of female C. nebulosa in part i. (c) The part ii of the male C. nebulosa suddenly becomes shorter. (d) The part ii of female C. nebulosa same as male. (e) The part iii of male C. nebulosa. (Inside circle is the trichoid sensilla). (f) The part iii of female *C. nebulosa*. (g) The part iv of male *C. nebulosa*. (h) The part iii of female *C. nebulosa*. (i) The part iv of male *C. nebulosa*. (j) The part v of female *C. nebulosa*. Inside circle is the trichoid sensilla and grooved peg sensilla in the box.

similar pattern of sensilla distribution in the part v (Fig 7I and 7J). So, the sexual dimorphism in the sensillar distribution observed in sexes is not evident in the part v.

## 4. Discussion

In this publication, we provide the first detailed description of the ultrastructure of the external morphology and distribution of antennal sensilla in both males and females of *C. nebulosa*.

We illustrated that the antennae exhibiting sexual dimorphism in the number of grooved peg sensilla on the flagellum as well as the length of the antennae. If we compared to *C. nebulosa* with *T. aridifolia*, both of them possess seven types of sensilla, but display some differences in their distribution. Additional study is needed to determine if this is species-, genus-, or family-level differences.

## 4.1 Structural features and the putative function of each type of sensilla

Grooved peg sensilla was reported in numerous insects [17–19]. In previous studies of mantis, this type of sensilla was referred either as "basiconic sensillum" [13,20] or "medium and short thin-walled peg sensillum" [21]. Based on transmission electron microscopy, we preferred the name "grooved peg sensilla", the double-walled cuticular wall with external grooves and the un-branching sensory cilia correspond to grooved peg sensilla. [16]. Sex-related differences in the number of grooved peg sensilla were previously observed in *Anagyrus vladimiri* [22]. Grooved peg sensilla are significantly more numerous in males antennae compared to females of *C. nebulosa*. This trait is related to the need for the males to detect sex pheromones in the air. They believed to serve as olfactory, hygro-, thermo-, or mechano-receptors and they can sense the $CO_2$ concentration [23].

The basiconic sensilla have a non-flexible socket, the base contains two sensory neurons and the cilia branched into two to several branches in the hair lumen of the sensilla [24–27]. In many species of Hymenoptera and Diptera, it is distributed in the pedicel and flagellomeres. Functionally, in fruit flies, single-walled and perforated basiconic sensilla receive signals from odor molecules belonging to alcohols and terpenes [28]. In the wild, these odorant molecules are the main components of plants, such as fruits and flowers. Electrophysiological experiments from this type of sensilla have revealed that the sensory neurons respond to odors of terpenes, alcohol, and banana [16]. Thus, the basiconic sensilla might be specialized to detect odors of plants, and a sparse distribution of this type sensilla might reflect predatory nature of mantids, which are less relying on plants as a food source. This idea agrees with the fact that this type of sensilla is absent in mosquitoes [29], but it is present in large numbers in cockroaches, locusts, and fruit flies [30–32], which are polyphagous and plant-feeding insects.

In *C. nebulosa*, trichoid sensilla are numerous. We did not observe any significant differences between male and female. Trichoid sensilla are recognized as olfactory organs, which are well conserved among the insect species from fruit flies to cockroaches [30–33]. They are considered to function as primary sex pheromone detectors, but some trichoid sensilla distributed in the scapus and pedicels is considered mechanoreceptive sensilla [24,34–36]. Trichoid sensilla of *C. nebulosa* are mainly present in the distal part of flagellum after about 15th flagellomere. This species has longer trichoid sensilla in males, and used to detect efficiently female-emitted sex pheromones. In addition, the length of trichoid sensilla increased from the proximal to the distal region of the antenna in both sexes. In some insect species (for example, *Culex quinquefasciatus Say*, *Aedes aegypti* and *Aphid Semiaphis heraclei*), both sexes have the trichoid sensilla of different lengths [37–39]. This difference is related to maturation of the sensilla and the developmental process of mantis antennae [40]. Difference in length of trichoid sensilla is observed within the same individual and between the sexes in *Helicoverpa armigera* [41]. In *C. nebulosa*, we did not observed any significant differences between the sexes.

The campaniform sensilla are usually found at the base of scape and pedicel, for example, true flies (Diptera), in that study, the diameter and number of campaniform sensilla was similar in both sexes [42]. This type of sensillum is thought to detect general stresses in the cuticle, possibly with some directional sensitivities [43–45]. In some Hemiptera insects, such as *Phenacoccus solenopsis* Tinsley, the campaniform sensilla were distributed only on male antennae,

indicating these sensilla are important for recognition of female sex pheromones [46]. The sensilla of the same type have different functions in other groups of insects.

Chaetic sensilla were reported in Hemiptera and Diptera. They are the most widely distributed sensilla on antennae of *C. nebulosa*, occurring on the scape, pedicel, and all flagellomeres. The chaetic sensilla are circularly distributed in a single line on each flagellomere in mantis with a strong and long-grooved cuticular apparatus and a flexible socket. According to their external morphology and distribution, the chaetic sensilla of scape and pedicel most likely have the mechanoreceptive function. The abundance of chaetic sensilla at the antennal tip is important for mechanically sensing distant objects. In other insects which have the same structure, it is considered to have mechano- and chemo-sensitive functions [47]. This sensilla are acting as contact chemoreceptors due to its presence at the apex pore, which is probably involved in host recognition and host acceptance [48].

Coeloconic sensilla are present on the flagellomeres in many insects and have various shapes. The coeloconic sensilla possess a grooved cuticular apparatus protruding from a central hole that stands in a round depression of 7 mm in diameter rimmed by a cuticular edge, which previously was described as a pit organ, since they are found within deep pits in *Augochloru puru* Smith (Hymenoptera: Apoidea) [49]. They were the least abundant sensilla in *C. nebulosa* and it is also found in low numbers (usually two or three, up to five or six) in many other insects, but the coeloconic sensilla are extremely abundant in the *Bittacus sinicus* Issiki [50]. The coeloconic sensilla mainly function as hygro- and thermo-receptors, which respond to changes in the ambient humidity and temperature, although they may have olfactory function to plant volatile, or are sensitive to $CO_2$ [51].

## 4.2 Comparison of antennae of *C. nebulosa* and *T. aridifolia*

In the previous research [19], the details of the sensillum types and their distribution in *T. aridifolia* were discussed. Comparing the antennae of the *C. nebulosa* and the *T. aridifolia*, we observed some differences in the external morphology of males, while the females have no significant differences except for length of antenna. In males of *C. nebulosa*, the distal part of each flagellomere is enlarged and the base is narrower (Figs 1, 4C, 4D, 7G and 7I), while the flagellomeres of *T. aridifolia* are almost equal in width through the length [21]. The flagellomeres also have equal width through their length in females of both species. The overall length of antenna of *T. aridifolia* is longer than that of *C. nebulosa*; the former also has the larger number of flagellomeres.

Based on distribution of sensilla, Carle and Toh *et al.* [21] divided the antennal flagellum of *T. aridifolia* into six parts. In this study, we described five distinct parts in the antennal flagellum of *C. nebulosa*. The primary difference between two species is the distribution of grooved peg sensilla. In males of *C. nebulosa*, the sensilla are present in the part i of the flagellum (Fig 4A–4D); while in *T. aridifolia*, only chaetic sensilla are present in the part i in male, and the grooved peg sensilla start to appear in the part ii. The closer to the distal part of flagellum, the more grooved peg sensilla could be observed. We speculate that the difference in the number and distribution of grooved peg sensilla may reflect the differences in living environment of *T. aridifolia* and *C. nebulosa*. *T. aridifolia* is a common species, inhabiting the open spaces of gardens, low shrubs, grass, it is also common in the low altitude of the mountains. *C. nebulosa* prefers the tropical rainforests with warm and humid climate. *C. nebulosa* living environment is more complex. In the long process of evolution, the species developed specialized sensilla responding to diverse stimuli coming from surrounding environment. In addition to the grooved peg sensilla, the trichoid sensilla are more abandon on the flagellum of the *C. nebulosa* compared to *T. aridifolia*.

Comparing the coeloconic sensilla of the *T. aridifolia* and *C. nebulosa*, we discovered a new subtype of the coeloconic sensilla of the later species. In the females of *C. nebulosa*, we found a hole at the distal end of the flagellum, and there was a structure in the hole similar to the central protrusion of the coeloconic sensilla, but it was thicker than common coeloconic sensilla on the antennae of praying mantis. The coeloconic sensilla has an external structure of hygroreceptive sensilla. The presence of these sensilla may also reflect the differences in the living condition of *C. nebulosa*. This subtype of sensilla was only observed on antennae of individual specimens, which could be accidental. We expect more detailed comparative study of different species of praying mantis.

## Supporting information

**S1 File.**
(PDF)

## Author Contributions

**Conceptualization:** Yuchen Wang, Yang Wang.

**Methodology:** Yuchen Wang, Yang Liu.

**Supervision:** Yang Wang, Peng Zhao.

**Writing – original draft:** Yuchen Wang, Tao Wan.

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
