## [Decision Letter · Decision Letter 0]

26 Oct 2023

PONE-D-23-30297Ultrastructure of the antennal sensilla in the praying mantis  Creobroter nebulosa  Zheng (Mantedea: Hymenopodidae)PLOS ONE

Dear Dr. Wang,

Thank you for submitting your manuscript to PLOS ONE. After careful consideration, we feel that it has merit but does not fully meet PLOS ONE’s publication criteria as it currently stands. Therefore, we invite you to submit a revised version of the manuscript that addresses the points raised during the review process.

We look forward to receiving your revised manuscript.

Kind regards,

Peng He, Ph.D

Academic Editor

PLOS ONE

Journal Requirements:

4. Thank you for stating the following financial disclosure: "National Natural Science Foundation of China (Grant no. 32200380); Natural science basic research project of Shaanxi Province (Grant No. 2020JQ-904)."

5. Thank you for stating the following in your Competing Interests section: "NO authors have competing interests".

Please complete your Competing Interests on the online submission form to state any Competing Interests. If you have no competing interests, please state "The authors have declared that no competing interests exist.", as detailed online in our guide for authors at http://journals.plos.org/plosone/s/submit-now.

Additional Editor Comments:

Dear Dr. Yang:

Your manuscript was reviewed carefully by myself and referred to reviewers for comment. On the basis of this study, I have decided that the manuscript is not acceptable in its present form and requires major revision before it can be considered for publication. Reviewer comments appear at the end of this letter.

We do not insist that you adopt all reviewer suggestions, but we do ask that you consider each carefully and explain your reasons for rejecting any one of them. Even if you feel that no change is justified for a specific point, please indicate your reason for not agreeing with the reviewer.

Reviewers' comments:

Reviewer's Responses to Questions

**Comments to the Author**

1. Is the manuscript technically sound, and do the data support the conclusions?

Reviewer #1: Partly

Reviewer #2: Yes

2. Has the statistical analysis been performed appropriately and rigorously? 

Reviewer #1: N/A

Reviewer #2: No

3. Have the authors made all data underlying the findings in their manuscript fully available?

Reviewer #1: Yes

Reviewer #2: Yes

4. Is the manuscript presented in an intelligible fashion and written in standard English?

Reviewer #1: No

Reviewer #2: Yes

5. Review Comments to the Author

Reviewer #1: The manuscript must be improved to be published. The title of the paper should correct to Ultrastructure of the antennal sensilla of the praying mantis Creobroter nebulosa Zheng (Mantedea: Hymenopodidae)". Abstract must improve and highlights on the aim and findings should added in clear order.

Revision of the English would be greatly appreciated (I already made some). Some sentences are unclear because of errors in English style and grammar.

In introduction section, updated citations should include (I add some) and generally needs improvement. In materials and methods section some details is missed that must be clear (E.g. how authors differentiate between the adult of both sex???).

In results section, improvement in writing is a demand especially Page 12. Some pictures need higher magnification that would be of help and Authors must add scale bars to inset images. Wrong identification for sensillum types that must be corrected. For examples:

Fig. 2

It is Bohm bristles not basiconic sensilla

Magnified view for SCh should added.

Fig. 3

Missed labelling for sensillum types and magnified of inset micrographs for each sensillum type should include.

Fig 4

magnified of inset micrographs for each sensillum type should include.

Fig. 6

B and C: Arrows and labelling for the mentioned sensillum types must added, for example B containing ST and basiconic type

e and f:

Both micrographs are the same sensilla campaniform. Minor difference around it is only according to the case of external stress on the antenna

Fif 7:

Wrong identification in C and D micrograph: how did authors identify both as the same type >> it is clearly different in size even you see the magnification bar.

F and H is sensillum coeloconica not coelocapitular.

Discussion section is rather weak and must be modify. I am of opinion that discussion section could be significantly improved. Right now I found it a bit synthetic. Would like to see more discussion on how glands/sensilla different between species, especially of different Hemiptera families. Moreover, you should improve discussion dealing with glands/sensilla ultrastructural organization and give more attention to ecological role these glands could have.

Detailed comments were explored in the attached file with tracking mode that could be shared with the authors where I detail my issues with the MS.

Reviewer #2: Wang et al publication includes potential interest and the main topic that they authors addressed is sound to me. They investigated the detailed fine morphological of the

antennae of the praying mantis Creobroter nebulosa using light and scanning electron microscopy (SEM) in both males and females. The authors discussed the difference of antennal sensilla between C. nebulosa and Tenodera aridifolia Stoll as an example of a sit-and-wait predator. The manuscript is well written and easy to be followed. I suggest the manuscript can be published in Plos One after minor revision. Several suggestions and comments are listed below which need to be addressed before publication:

Comments:

1- All figures need to be improved.

2- The authors should describe the rearing conditions of species under study.

3- Authors should report statistical methods in sufficient detail.

4- Authors should cite the following reference in the (Introduction line 42) as mentioned below:

Abd El-Ghany NM, Zhou JJ, Dewer Y. Antennal sensory structures of Phenacoccus solenopsis (Hemiptera: Pseudococcidae). Front Zool. 2022 Dec 15;19(1):33. doi: 10.1186/s12983-022-00479-4.

5- Female antenna in figure (1-b) needs to be re-captured.

6. PLOS authors have the option to publish the peer review history of their article (what does this mean?). If published, this will include your full peer review and any attached files.

Reviewer #1: No

Reviewer #2: No

---

## [Author Response · Author response to Decision Letter 0]

12 Dec 2023

We have made changes as requested by the editor. The reviewers' response to the manuscript has been detailed and uploaded in the "Response to Reviewers"

---

## [Decision Letter · Decision Letter 1]

23 Feb 2024

PONE-D-23-30297R1Ultrastructure of the antennal sensilla of the praying mantis  Creobroter nebulosa  Zheng (Mantedea: Hymenopodidae)PLOS ONE

Dear Dr. Wang,

Thank you for submitting your manuscript to PLOS ONE. After careful consideration, we feel that it has merit but does not fully meet PLOS ONE’s publication criteria as it currently stands. Therefore, we invite you to submit a revised version of the manuscript that addresses the points raised during the review process.

We look forward to receiving your revised manuscript.

Kind regards,

Peng He, Ph.D

Academic Editor

PLOS ONE

Reviewers' comments:

Reviewer's Responses to Questions

**Comments to the Author**

1. If the authors have adequately addressed your comments raised in a previous round of review and you feel that this manuscript is now acceptable for publication, you may indicate that here to bypass the “Comments to the Author” section, enter your conflict of interest statement in the “Confidential to Editor” section, and submit your "Accept" recommendation.

Reviewer #1: (No Response)

2. Is the manuscript technically sound, and do the data support the conclusions?

Reviewer #1: Partly

3. Has the statistical analysis been performed appropriately and rigorously? 

Reviewer #1: Yes

4. Have the authors made all data underlying the findings in their manuscript fully available?

Reviewer #1: Yes

5. Is the manuscript presented in an intelligible fashion and written in standard English?

Reviewer #1: No

6. Review Comments to the Author

Reviewer #1: Revision of the English would be greatly appreciated. Some sentences are too long , or unclear because of errors in English style and grammar. Abstract, introduction, and materials and methods sections need improvement in some parts. Results section still need improvement and considering the subsequent numbering for figures (mentioned in PDF file). Discussion section is rather need effort to be more reliable. Finally, references list must be checked carfully (excess citations:75 ref. and contains non-relivant references that I suggest deletion)

Detailed comments were explored in the attached file that could be shared with the authors where I detail my issues with the MS.

7. PLOS authors have the option to publish the peer review history of their article (what does this mean?). If published, this will include your full peer review and any attached files.

Reviewer #1: No

---

## [Author Response · Author response to Decision Letter 1]

11 Mar 2024

Thanks for your efforts on our manuscript, especially for the academic advice and language improvement. We streamlined and refined the language to improve readability. We also involved a native English-speaking entomologist for language corrections

.The highlighted part has been revised according to your comments, the specific modifications are presented in ‘Revised Manuscript with Track Changes’. We checked the references carefully and cut them down to 51.

In addition, Tao Wan participated in the manuscript revision, including some pictures rephotography and writing. So we add Wan Tao as one of the authors.

We would like to express our great appreciation to you and reviewers for comments on our paper. Looking forward to hearing from you.

---

## [Editor Report · Decision Letter 2]

17 Mar 2024

Ultrastructure of the antennal sensilla of the praying mantis  Creobroter nebulosa  Zheng (Mantedea: Hymenopodidae)

PONE-D-23-30297R2

Dear Dr. Wang,

We’re pleased to inform you that your manuscript has been judged scientifically suitable for publication and will be formally accepted for publication once it meets all outstanding technical requirements.

Kind regards,

Peng He, Ph.D

Academic Editor

PLOS ONE
---

## [Editor Report · Acceptance letter]

9 May 2024

PONE-D-23-30297R2 

PLOS ONE

Dear Dr. Wang, 

I'm pleased to inform you that your manuscript has been deemed suitable for publication in PLOS ONE. Congratulations! Your manuscript is now being handed over to our production team.

Kind regards, 

on behalf of

Dr. Peng He 

Academic Editor

PLOS ONE